# *Parkia platycephala* Lectin (PPL) Inhibits Orofacial Nociception Responses via TRPV1 Modulation

**DOI:** 10.3390/molecules27217506

**Published:** 2022-11-03

**Authors:** Gerlânia de Oliveira Leite, Sacha Aubrey Alves Rodrigues Santos, Romério Rodrigues dos Santos Silva, Claudener Souza Teixeira, Adriana Rolim Campos

**Affiliations:** 1Núcleo de Biologia Experimental, Universidade de Fortaleza, Fortaleza 60000, Brazil; 2Departamento de Bioquímica e Biologia Molecular, Universidade Federal do Ceará, Fortaleza 60000, Brazil; 3Centro de Ciências Agrárias e da Biodiversidade, Universidade Federal do Cariri, Crato 63100, Brazil

**Keywords:** *Parkia platycephala*, lectins, Orofacial Nociception

## Abstract

Lectins are a heterogeneous group of proteins that reversibly bind to simple sugars or complex carbohydrates. The plant lectin purified from the seed of *Parkia platycephala* (PPL) was studied. This study aimed to investigate the possible orofacial antinociceptive of PPL lectin in adult zebrafish and rodents. Acute nociception was induced by cinnamaldehyde (0.66 μg/mL), 0.1% acidified saline, glutamate (12.5 µM) or hypertonic saline (5 M NaCl) applied into the upper lip (5.0 µL) of adult wild zebrafish. Zebrafish were pretreated by intraperitoneal injection (20 µL) with vehicle (Control) or PPL (0.025; 0.05 or 0.1 mg/mL) 30 min before induction. The effect of PPL on zebrafish locomotor behaviour was evaluated in the open field test. Naive groups were included in all tests. In one experiment, animals were pre-treated with capsazepine to investigate the mechanism of antinociception. The involvement of central afferent C-fibres was also investigated. In another experiment, rats pre-treated with PPL or saline were submitted to the temporomandibular joint formalin test. Other groups of rats were submitted to infraorbital nerve transection to induce chronic pain, followed by induction of mechanical sensitivity using von Frey. PPL reduced nociceptive behaviour in adult zebrafish, and this is related to the activation of the TRPV1 channels since antinociception was effectively inhibited by capsazepine and by capsaicin-induced desensitization. PPL reduced nociceptive behaviour associated with temporomandibular joint and neuropathic pain. The results confirm the potential pharmacological relevance of PPL as an inhibitor of orofacial nociception in acute and chronic pain.

## 1. Introduction

According to the International Association for the Study of Pain (IASP, 2008), pain is characterised as an unpleasant sensory and emotional experience, associated with actual or potential tissue damage, or described in terms of this damage. The damage is one of the classic signs of the inflammatory process and initially results from the sensitization of nociceptors, besides being present as a symptom of many clinical disorders that affect a large portion of the population, causing damage to the quality of life [1,2]. Natural molecules have a great biotechnological potential and arouse great scientific interest; among these, the lectins are a subject of growing pharmacological interest, highlighting the antinociceptive studies [3,4,5].

Lectins are proteins of non-immunological origin with the ability to recognize and bind to carbohydrates and glycoconjugates [6]. The most studied lectins are those of plant origin, and these proteins have some biological applications already described, such as antitumor [7], antidepressant [8], anti-inflammatory [9] and antinociceptive activity [3,4,5].

In general, the biological activities described for lectins are related to the ability of these proteins to recognize carbohydrates on the surface of cell membranes and trigger an intracellular signalling that results in a biological response [10]. 

Recently, our research group described the first work showing the effect of a lectin in inhibiting orofacial pain in a zebrafish model [4]. Subsequently, it was shown that the antinociceptive effect is related to the mode of interaction of lectins with cell carbohydrates [4]. The zebrafish has become a prominent vertebrate model for the study of diseases and new drug discovery [11] based on its phenotypes, and has been widely used for pharmacological studies [12]. The zebrafish has had its genome sequenced, which revealed that this experimental model has highly conserved genes that participate in oxidation, lipid metabolism and inflammation, and many of the proteins in this small animal appear to function in a mammalian-like manner [13,14].

Species of the genus Parkia are used in traditional medicine as an analgesic drug, especially against dental pain [15]. *Parkia platycephala* is a species found predominantly in the North and Northeast of Brazil and is popularly known as visgueiro or faveira [16]. *Parkia platycephala* glucose/mannose lectin (PPL) was the first of its genus to have its amino acid sequence elucidated. The primary structure is formed by 447 amino acid residues and the three-dimensional structure is formed by three β-prism domains, and each one has a non-identical mannose binding site [17,18]. PPL has some biological activities described, of which we highlight its parasitic activity and modulation of antibiotic activity [19], antinociceptive and anti-inflammatory activity with acetic acid and peritonitis -induced nociception model [20].

This study aimed to investigate the possible orofacial antinociceptive of *Parkia platycephala* (PPL) lectin in adult zebrafish and rodents.

## 2. Results

### 2.1. Lectin Purification

The purification of PPL was achieved by affinity chromatography on sephadex-G100 column, as previously described by Cavada et al. [17]. The chromatogram profile showed two peaks, the first (PI) corresponding to unbound protein fraction and the second (PII) corresponding to the retained protein fraction (Figure 1A). In gel electrophoresis under denaturing conditions (SDS-PAGE), PII showed a single band with a relative mass of 50 kDa, which corresponds to molecular weight of the PPL (Figure 1B).

### 2.2. Orofacial and Corneal Antinociceptive Activity in Adult Zebrafish

Capsaicin injection into the lips of adult zebrafish induced nociceptive behaviour (Figure 2A). PPL (0.025 mg/mL) reduced this behaviour (* *p* < 0.05 *vs* Control) and this effect was prevented by the pre-treatment with capsazepine (Figure 2A). PPL at all doses tested had a significant effect (* *p* < 0.05–** *p* < 0.01 *vs* Control) on the nociceptive behaviour evoked by the local application of 5 M NaCl solution to the corneal surface (Figure 2B).

According to Figure 3, the depletion of TRPV1+ C-fibres by capsaicin decreased the glutamate-induced orofacial nociceptive activity (**** *p* < 0.001 *vs* control) and abolished the antinociceptive effect of PPL (0.025 mg/mL).

### 2.3. Capsaicin-Induced Orofacial Nociceptive Behaviour in Mice

Capsaicin injection into the upper lip of mice induced face rubbing (Figure 4A). Pre-treatment with PPL was associated with a reduction in face-rubbing behaviour induced by the application of capsaicin (**** *p* < 0.001 *vs* Control).

### 2.4. Formalin-Induced Temporomandibular Nociception in Rats

The injection of formalin into the temporomandibular junction induced the nociceptive behaviours of head flinching, face rubbing and chewing. As shown in Figure 4B, pre-treatment with PPL (0.25 mg/kg), but not a control vehicle, reduced the formalin-induced behaviours (*** *p* < 0.01 *vs* Control). 

### 2.5. Neuropathic Orofacial Nociception Induced by Infraorbital Nerve Transection

Left IONX produced sustained hypersensitivity to facial mechanical stimulation that was reflected in a reduced mechanical withdrawal threshold for 21 days. The thresholds in sham-operated and naive rats did not significantly change. To investigate the effects of PPL on the mechanical withdrawal threshold in IONX rats, PPL (0.25 mg/kg) or vehicle (control) was administrated at post-operative days 1, 3, 5, 7, 10, 14 and 21. PPL reversed the reduced mechanical threshold on post-operative days 3–21 when compared with Control (Figure 5).

## 3. Material and Methods

### 3.1. Animals

#### 3.1.1. Zebrafish

Adult wild zebrafish (*Danio rerio*) of both sexes (short-fin phenotype), aged 60–90 days, of a similar size (3.5 ± 0.5 cm) and weight (0.3 ± 0.2 g), were obtained from Agroquímica: Comércio de Produtos Veterinários LTDA (Fortaleza, Ceará, Brazil). Groups of 50 fish were acclimatized for 24 h in a 10-L glass tank (30 × 15 × 20 cm) containing dechlorinated tap water (ProtecPlus^®^) and an air pump with a submerged filter, at 25 °C and pH 7.0, under near-normal circadian rhythm (14:10 h of light/dark). The fish received *ad libitum* feed 24 h prior to the experiments. All experimental procedure was approved by the Ethics Committee on Animal Research of the Ceará State University (CEUA-UECE; #7210149/2016).

#### 3.1.2. Rodents

Swiss mice (20–30 g) and Wistar rats (250–300 g) obtained from the Núcleo de Biologia Experimental of the University of Fortaleza were used. They were housed (IVC cages, Tecniplast^®^) in environmentally controlled conditions (22 °C, 12 h light–dark cycle), with free access to standard pellet diet (Purina, São Paulo, Brazil) and water. The experimental protocols were approved by Animal Ethics Committee of the University of Fortaleza (CEUA-UNIFOR; 4250250817).

### 3.2. Lectin Purification

*P. platycephala* seeds were collected from plants located at Chapadinha, Maranhão, Brazil. Lectin purification was carried out by the affinity chromatography protocol, as previously described by Cavada et al. [17]. Mature seeds of *P. platycephala* were ground into a fine powder using a coffee mill. The flour was stirred with 0.15 M NaCl (1:10, m/v) for 4 h at 25 °C. The mixture was centrifuged at 10,000× *g* for 20 min at 4 °C. The clear supernatant (crude extract) was applied to a Sephadex G-100 column (2 × 10 cm) equilibrated with 150 mM NaCl containing. After washing of the unbound material in the equilibrium solution, the lectin was eluted with 100 mM mannose, pooled, exhaustively dialyzed against distilled water, and freeze-dried. Absorbance at 280 nm was used to estimate the protein concentration in all chromatographic fractions. The retained peak (*P. platycephala* lectin—PPL) was lyophilized and tested for purity by SDS-PAGE [18].

### 3.3. General Protocol

On the day of the experiment, zebrafish were randomly selected, then transferred to a wet sponge for treatment with the study lectin or controls, intraperitoneally (i.p.), after which they were placed in individual beakers (250 mL) containing 150 mL of water from the fish tank and allowed to recover. For oral treatments, a 20 μL variable pipette was used. An insulin syringe (UltraFine^®^ BD) with a 30-gauge needle was used to inject the lips or for the intraperitoneal treatment.

PPL was administered to zebrafish (20 μL, i.p.) at 0.025; 0.05 or 0.1 mg/mL [5]. Mice received PPL at doses of 0.25, 0.5 or 1 mg/kg (10 mL/kg, i.p.), and rats received PPL at a dose of 0.25 mg/kg (10 mL/kg, i.p.).

### 3.4. Orofacial Antinociceptive Activity in Adult Zebrafish

The concentrations of the noxious and antagonistic agents, as well as the time of nociceptive action analysis used, were based on our previous reports: 0.66 μg/mL cinnamaldehyde [21], 0.1 % acidified saline [22] and 12.5 mM glutamate [23]. After treatment and application of the algogenic agent, the animals were then placed in a glass Petri dish (∅ 15 cm), divided into quadrants, and the nociceptive response was quantified in terms of locomotor activity performed during a certain time, specific for each model described below.

#### 3.4.1. Capsaicin-Induced Orofacial Nociceptive Behaviour

Orofacial nociception was induced with capsaicin (TRPV1 agonist; 40.93 μM; 5.0 μL) dissolved in ethanol, PBS, and distilled water (1:1:8), injected into the lips of the animals (*n* = 8/group), 30 min after pre-treatment (20 μL) with PPL (0.025 mg/mL) or vehicle (Control, saline). Another group (*n* = 8) received capsazepine (TRPV1 antagonist; 0.5 mg/mL) 15 min before PPL. A naive group (*n* = 8) was included. The antinociceptive activity was observed individually during 10–20 min.

#### 3.4.2. Evaluation of the Involvement of Central Afferent Fibres to Capsaicin

Repeated treatment with interval capsaicin produces a reduction in functional response, causing desensitization of type C afferent fibres [24]. In this study, we used the method proposed by Soares et al. [25]. Adult zebrafish (*n* = 8/group) were grouped into: 

(a) Control-saline (20 μL; i.p.).

(b) Desensitized Control (DC)-capsaicin (40.93 μM; 20 μL; i.p.) administered at times 15, 30, 60, 120 and 240 min before saline.

(c) PPL (0.025 mg/mL; 20 μL; i.p,).

(d) PPL (0.025 mg/mL; desensitized)-capsaicin (40.93 μM; 20 μL; i.p.) administered at times 15, 30, 60, 120 and 240 min before PPL.

(e) Naive.

Orofacial nociception was induced with glutamate, as described in Section 2.3.

#### 3.4.3. Corneal Nociception Model in Adult Zebrafish

Corneal nociception was induced with hypertonic saline (5 M NaCl; 5.0 μL) injected into the right eye of the animals (*n* = 8/group), 30 min after pre-treatment with PPL (see 3.2) or vehicle (Control, saline; 20 μL). A naive group (*n* = 8) was included. The animals were then placed in a glass Petri dish (∅ 15 cm), divided into quadrants and the nociceptive response was quantified in terms of locomotor activity performed during 0–5 min.

### 3.5. Capsaicin-Induced Orofacial Nociceptive Behaviour in Mice

Orofacial nociception was induced by subcutaneous application o (*n* = 6/group), 30 min after pre-treatment with PPL or vehicle (Control, saline; 10 mL/kg). A naive group (*n* = 6) was included. The mice were observed in related capsaicin (40.93 μM; 20 μL) into the perinasal area (vibrissae) of the right cheek of the animals to the time(s) of facial rubbing behaviour with the posterior and/or anterior ipsilateral paws during the period 10–20 min, as described by Pelissier et al. [26].

### 3.6. Formalin-Induced Temporomandibular Nociception in Rats

Rats (*n* = 6/group) were acclimated individually in a glass test chamber (30 × 30 × 30 cm) for 30 min to minimize stress. The animals were pre-treated (10 mL/kg; i.p.) with PPL (0.25 mg/kg) or vehicle (Control, saline) and, 30 min later, the animals were anesthetized (i.p.) with ketamine (100 mg/kg) and xylazine (10 mg/kg), and the left temporomandibular joint was injected with 50 μL of 2 % formalin. A sham group receiving 0.9 % NaCl (50 μL; left temporomandibular joint) and a naive group were also included (*n* = 6/group) [27].

After their recovery from anaesthesia (20 min), the animals were returned individually to the test chamber to quantify nociception as asymmetrical rubbing of the orofacial region with the ipsilateral fore- or hind-paw, and as head flinching (intermittent and reflexive shaking of the head) [28]. The time that the rat spent rubbing the orofacial region was registered 12 times at 3 min intervals; head flinching was registered by its absence or presence.

### 3.7. Neuropathic Orofacial Nociception Induced by Infraorbital Nerve Transection

Rats were anesthetized (i.p.) with ketamine (100 mg/kg) and xylazine (10 mg/kg) to expose the left infraorbital nerve (ION) at its entry into the infraorbital foramen by way of an intra-oral incision (2 mm) in the oral mucosa of the left fronto-lateral maxillary vestibulum, as previously described [15]. The ION was lifted from the maxillary bone and cut (IONX) without damaging adjacent nerves and vessels. Subsequently, the animals were returned to their cages and fed with mash and chow. The animals were monitored daily in the post-operative period. Rats were divided into two groups (*n* = 6/group): PPL (0.25 mg/kg; i.p.) or vehicle (10 mL/kg saline; i.p.; Control). Naive and sham-operated animals (*n* = 6/group) were used as controls.

The rats were acclimated, trained and tested for facial mechanical sensitivity (head withdrawal threshold) 1 day prior to nerve transection (baseline) and on post-operative days 1, 3, 5, 7, 10, 14 and 21, as previously described [15]. A single dose of PPL or vehicle (*n* = 6/group) was administered on each post-operative day. Mechanical sensitivity of the left whisker pad skin was assessed using an electronic von Frey instrument at 30 min after the PPL or vehicle treatment. The head-withdrawal threshold for mechanical stimulation of the whisker pad skin was defined as the minimum force needed to evoke an escape more than three times as a result of five stimuli. 

### 3.8. Statistical Analyses

The results are presented as mean ± S.E.M. of each group of animals. Normality of distribution was confirmed (Kolmogorov-Smirnov) and data were submitted for the analysis of variance (one-way ANOVA), followed by the Tukey post-hoc test, or two-way ANOVA, followed by Bonferroni’s post-hoc test. For analyses of PPL (and vehicle) effects in IONX animals, the baseline was normalized, and all values post-PPL or post-vehicle administration were expressed as percentage (%) change from baseline. The level of statistical significance was set at 5% (*p* < 0.05).

## 4. Discussion

In our previous work, we verified that PPL produced an orofacial antinociceptive effect in adult zebrafish, and this activity is related to the interaction of PPL with glycans present in the TRPV1 receptor [5]. In the present study, we deepened knowledge about the participation of the TRPV1 channel in the orofacial antinociceptive effect of PPL (mannose/galactose-binding lectin), as well as evaluated if this effect would also be verified in rodents.

PPL, as expected, reduced nociceptive behaviour induced by capsaicin applied to the lip. Besides this, PPL prevented nociceptive behaviour induced by hypertonic saline applied into the cornea. Since the hypertonic saline stimulus in the human cornea is related to the activation of the TRPV1 channels [29], this result support the idea that PPL acts on the TRPV1 receptor.

To investigate the modulation of TRPV1 receptors by PPL, adult zebrafish were pre-treated with the TRPV1 receptor antagonist capsazepine. This compound prevented the antinociceptive effect of PPL. Besides this, depletion of TRPV1+ C fibres by capsaicin eliminated antinociception produced by PPL. Soares et al. [25] demonstrated that the progressive and intermittent use of capsaicin led to an antinociceptive response, and this effect is blocked by oleanolic acid, corroborating the understanding that PPL is a potential drug acting on this pathway. Since we found PPL exhibited orofacial antinociceptive activity in adult zebrafish, we next examined its activity in rodents. 

TRPV1 is a glycosylated receptor composed of N-glycans rich in mannose residues. Veldhuis and collaborators [30] demonstrated that the glycosylation of TRPV1 is a determinant for capsaicin-evoked desensitization and ion permeability. Plant lectins can recognize and interact with N-glycans from different cells, eliciting cellular responses that characterize some biological activities, of which we highlight inhibition of viral infection [31], antitumor activity [32], immunomodulatory activity [33] and anti-inflammatory activity [4]. Therefore, we hypothesized that the antinociceptive effect of PPL in inhibiting capsaicin activity may be related to the interaction of PPL with mannose residues present in the N-glycans of the TRPV1 receptor.

Temporomandibular disorders are musculoskeletal pain conditions characterized by pain in the temporomandibular joint and/or masticatory muscles [34]. Here, temporomandibular joint nociception induced by formalin was prevented by PPL, thus suggesting its use for the treatment of this disorder. 

Receptors are up-regulated after the injury of the infraorbital nerve and to the fact that pharmacological blockade or selective ablation of trigeminal ganglia cells expressing TRPV1 results impaired the development of facial hyperalgesia [35,36]. Using this rodent model of facial neuropathic pain, we found that treatment with PPL could reduce facial mechanically induced hypersensitivity. This effect may be due to an antagonistic action of PPL on TRPV1 channels.

In conclusion, our results indicate that PPL might to be an effective analgesic drug in acute and chronic orofacial pain states and that this effect may be due to the modulation of the TRPV1 channel. 

## Figures and Tables

**Figure 1 molecules-27-07506-f001:**
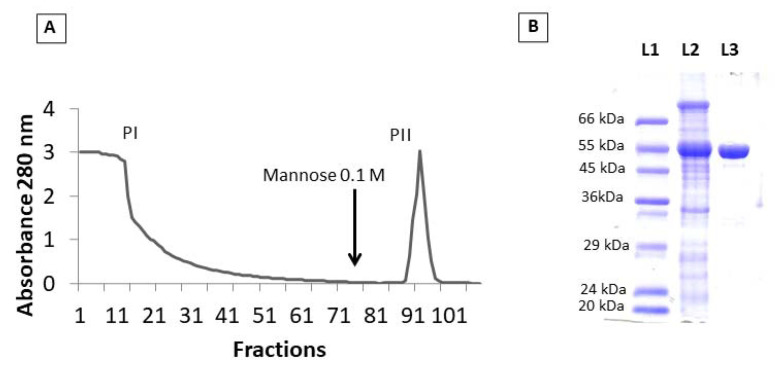
PPL purification. (**A**) Elution profile of PPL in Sephadex G-100 chromatography. (**B**) SDS–PAGE profile, (L1) Molecular mass markers: bovine serum albumin, 66 kDa; glutamic dehydrogenase, 55 kDa; ovalbumin, 45 kDa; glyceraldehyde-3-phosphate dehydrogenase, 36 kDa; carbonic anhydrase, 29 kDa; trypsinogen, 24 kDa and trypsin inhibitor, 20 kDa. (L2) Crude protein extract. (L3) PPL obtained in the PII of the Sephadex G-100.

**Figure 2 molecules-27-07506-f002:**
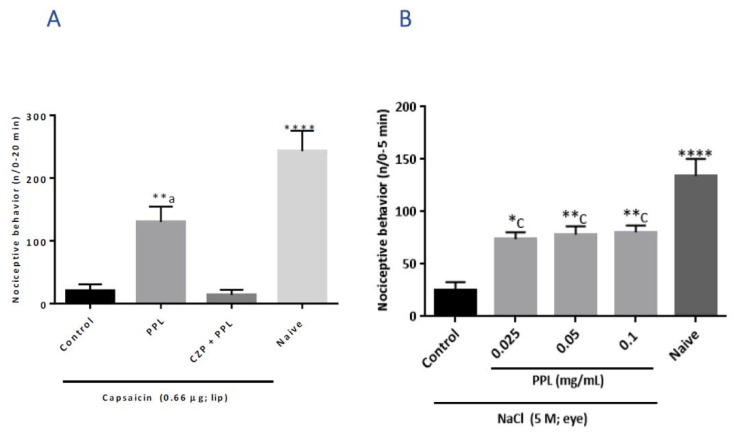
Effect of PPL on capsaicin-induced orofacial nociception (**A**) and on hypertonic saline-induced corneal nociception (**B**) in adult zebrafish. The results are expressed as mean values ± standard error of the number of nociceptive behaviours for each group. ANOVA followed by Tukey test. * *p* < 0.05, ** *p* < 0.01 *vs* Control; **** *p* < 0.0001 *vs* Control and CZP+PPL; ^a^
*p* < 0.01 *vs* CZP+PPL and Naive; ^c^
*p* < 0.01 *vs* Naive. CZP = capsazepine; PPL = 0.05 mg/mL PPL.

**Figure 3 molecules-27-07506-f003:**
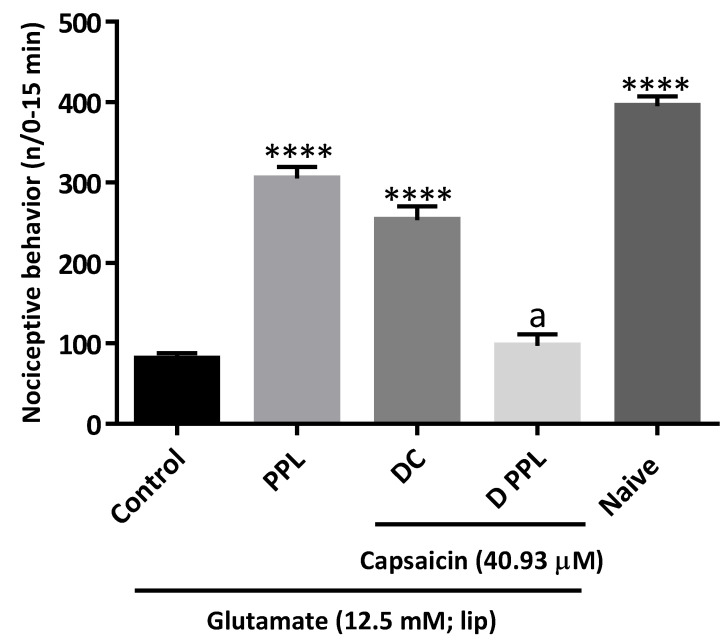
Involvement of the central afferent fibres sensitive to capsaicin (TRPV1) on the antinociceptive effect of PPL in adult zebrafish. Each column represents the mean ± S.E.M. (*n* = 6/group). ANOVA followed by Tukey test. **** *p* < 0.001 *vs* Control and ^a^
*p* < 0.0001 *vs* PPL, Control and Naive. DC = Desensitized Control; D PPL = Desensitized PPL (0.05 mg/mL); PPL = 0.05 mg/kg PPL.

**Figure 4 molecules-27-07506-f004:**
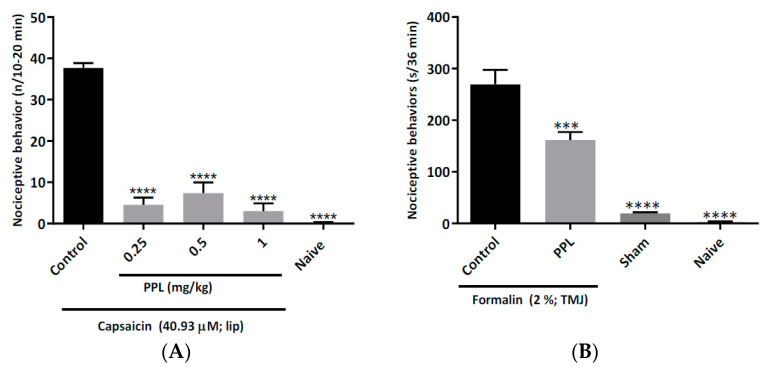
Effect of PPL on capsaicin-induced orofacial nociception in mice (**A**) and on formalin-induced temporomandibular junction nociception in rats (**B**). The results are expressed as mean values ± standard error of the number of nociceptive behaviours for each group. ANOVA followed by Tukey test. *** *p* < 0.01 and **** *p* < 0.0001 *vs* Control. PPL = 0.25 mg/kg PPL.

**Figure 5 molecules-27-07506-f005:**
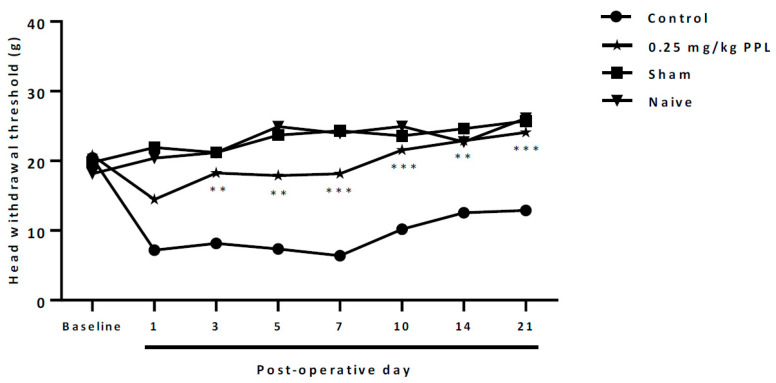
Antihyperalgesic effect of repeated PPL treatment after the infraorbital nerve transection. ANOVA followed by the Bonferroni test ** *p* < 0.01 and *** *p* < 0.001 vs Control. ANOVA followed by the Bonferroni test. PPL = 0.25 mg/kg PPL.

## Data Availability

Not applicable.

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
