# Peer review of "Parkia platycephala* Lectin (PPL) Inhibits Orofacial Nociception Responses via TRPV1 Modulation"

_molecules, 2022, doi:10.3390/molecules27217506_

Round 1

Reviewer 1 Report

In the manuscript, the authors evaluated the effect of a mannoside-specific legume lectin for its antinociceptive effect in acute and neuropathic orofacial nociception in zebrafish, mice and rats. The manuscript it very interesting and a nice contribution to the lectin field.

Comments:

1 - There are a few grammar mistakes and typos.

2 - Section 2.2 "Lectin purification" needs checking. It has been said that the resuspended fraction has been applied in the matrix, but I did not found any precipitation steps. Also, why the authors extracted the proteins at 37 degrees? 

3 - Did the authors compare the effect of the lectin with a known antinociceptive agent? 

4 - Does the carbohydrate-binding capacity of the lectin abolish/reduce the observed effect? 

5 - Figure 2 - the "micro" symbol of the figure is blurred.

Author Response

Response to Reviewer 1 Comments

Point 1: There are a few grammar mistakes and typos.

Response 1: Thanks for your comment. The text was reviewed.

Point 2 - Section 2.2 "Lectin purification" needs checking. It has been said that the resuspended fraction has been applied in the matrix, but I did not found any precipitation steps. Also, why the authors extracted the proteins at 37 degrees? 

Response 2: Thanks for your comment. The purification methodology was revised in the text.

Point 3 - Did the authors compare the effect of the lectin with a known antinociceptive agent? 

Response 3: Thanks for your comment. To contribute to the 3R principle, since our objective was mainly to deepen the understanding of the antinociceptive effect of PPL, we decided not to use a positive control group and thus reduce the number of animals used in the study.

Point 4 - Does the carbohydrate-binding capacity of the lectin abolish/reduce the observed effect? 

Response 4: Thanks for your comment. As mentioned in the first paragraph of our discussion and in previous works carried out by our group, we found that the carbohydrate binding capacity of the lectin cancels/reduces the observed effect, as this activity is related to the interaction of PPL with the glycans present at the TRPV1 receptor.

Point 5 - Figure 2 - the "micro" symbol of the figure is blurred.

Response 5: Figure 2 - the "micro" symbol of the figure was fixed.

Reviewer 2 Report

The article entitled “Parkia platycephala lectin (PPL) inhibits orofacial nociception response via TRPV1 modulation” investigates the effectiveness of purified lectin in formalin induced acute orafacial  noception.  The study has novelty and appropriately designed and executed in methodology related to animal experiments.  However, does not provide subsequent correlation with biochemical events and signaling mechnaism related to lectin mediated reductions in orofacial noceptions. It needs major and extensive revision of the article addressing the following comments. 

Comments:

·         The study had used the PPL lectin and had not provided sufficient information on lectin biological property, its anti inflammatory property and justification with relevant hypotheisis to use this lectin to investigate its efficacy in orofacial noceptions.  I think it needs to be added in the introduction portion. 

·         What about the purity of the lectin, In the protocol it was described to use SepahadexG100 affinity column provide the SDS-PAGE and state the purity of the lectin sample used in the study (Add at least in supplemtary figures).

·         The lectin sample was injected intraperitonially, how the dosing and concentration was decided for the experiments.  Provide information on the efficacy of lectin to mobilize and induce effect to orofacial system to reduce pain.   

·         The Line 113-115, The PPL is administered to zebrafish, mice and rat, all at single dose as it appears, No information on dosing pattern and periods.  No information on the toxicity, dose determination and other related information for all the three models.  Kindly revise the relevant sections and add detail description in methodology.

·         Is any information known about the antiinflamatory and analgesic property of purified PPL in any animal model?  Add the description in introduction with relevant references.

·         The article has to revise extensively for the English corrections and need to take acre on grammatical and typographical mistakes ex. Line 15 applied into in the upper lip need to be “applied into the upper lip”

·         The study interprets that lectin bind to TRPV1 receptor as it is glycosylated, and does not provide any relevant studies or reference that shows the PPL interaction with TRPV1 and evidence of downstream activation of ion channels for neuromodulations. 

The article had novelty and it requires proper correlation of the study to relate to the claimed outcome and need to revise extensively considering all these points.  If required can exhibit additional studies and correlate the outcomes of the lectin mediated efficacy in reducing orofacial noception.  

Round 2

Reviewer 2 Report

None